# Epidemiology, Treatment, and Prevention of Nosocomial Bacterial Pneumonia

**DOI:** 10.3390/jcm9010275

**Published:** 2020-01-19

**Authors:** Shio-Shin Jean, Yin-Chun Chang, Wei-Cheng Lin, Wen-Sen Lee, Po-Ren Hsueh, Chin-Wan Hsu

**Affiliations:** 1Department of Emergency, School of Medicine, College of Medicine, Taipei Medical University, Taipei 110, Taiwan; 101025@w.tmu.edu; 2Department of Emergency Medicine, Department of Emergency and Critical Care Medicine, Wan Fang Hospital, Taipei Medicine University, Taipei 110, Taiwan; 3Division of Thoracic Surgery, Department of Surgery, Wan Fang Hospital, Taipei Medical University, Taipei 110, Taiwan; 101403@w.tmu.edu.tw (Y.-C.C.); 101336@w.tmu.edu.tw (W.-C.L.); 4Division of Infectious Diseases, Department of Internal Medicine, Wan Fang Hospital, Taipei Medical University, Taipei 110, Taiwan; 89425@w.tmu.edu.tw; 5Department of Internal Medicine, School of Medicine, College of Medicine, Taipei Medical University, Taipei 110, Taiwan; 6Department of Laboratory Medicine, National Taiwan University Hospital, National Taiwan University College of Medicine, Taipei 100, Taiwan; hsporen@ntu.edu.tw; 7Department Internal Medicine, National Taiwan University Hospital, National Taiwan University College of Medicine, Taipei 100, Taiwan

**Keywords:** hospital-acquired pneumonia, ventilator-associated pneumonia, methicillin-resistant *Staphylococcus aureus*, *Pseudomonas aeruginosa*, extended-spectrum β-lactamase-producing Enterobacteriaceae species, extensively drug-resistant, *Acinetobacter baumannii* complex species, antibiotic combination, carbapenemase, ceftazidime-avibactam

## Abstract

Septicaemia likely results in high case-fatality rates in the present multidrug-resistant (MDR) era. Amongst them are hospital-acquired pneumonia (HAP) and ventilator-associated pneumonia (VAP), two frequent fatal septicaemic entities amongst hospitalised patients. We reviewed the PubMed database to identify the common organisms implicated in HAP/VAP, to explore the respective risk factors, and to find the appropriate antibiotic choice. Apart from methicillin-resistant *Staphylococcus aureus* and *Pseudomonas aeruginosa*, extended-spectrum β-lactamase-producing Enterobacteriaceae spp., MDR or extensively drug-resistant (XDR)-*Acinetobacter baumannii* complex spp., followed by *Stenotrophomonas maltophilia*, *Chryseobacterium indologenes*, and *Elizabethkingia meningoseptica* are ranked as the top Gram-negative bacteria (GNB) implicated in HAP/VAP. Carbapenem-resistant Enterobacteriaceae notably emerged as an important concern in HAP/VAP. The above-mentioned pathogens have respective risk factors involved in their acquisition. In the present XDR era, tigecycline, colistin, and ceftazidime-avibactam are antibiotics effective against the *Klebsiella pneumoniae* carbapenemase and oxacillinase producers amongst the Enterobacteriaceae isolates implicated in HAP/VAP. Antibiotic combination regimens are recommended in the treatment of MDR/XDR-*P. aeruginosa* or *A. baumannii* complex isolates. Some special patient populations need prolonged courses (>7-day) and/or a combination regimen of antibiotic therapy. Implementation of an antibiotic stewardship policy and the measures recommended by the United States (US) Institute for Healthcare were shown to decrease the incidence rates of HAP/VAP substantially.

## 1. Introduction

Patients with some underlying co-morbidities frequently need hospitalisation for various reasons. Once septicaemia occurs in these patients, they often experience prolonged hospital stays with increased healthcare expenses and high rates of mortality [1,2,3,4,5,6,7,8]. Amongst a variety of septicaemic entities, infection sites involving the lung parenchyma and/or complicated pleural empyema, intra-abdominal spaces, central nervous system, or unknown origins notably account for the leading causes of death [1,4,8,9,10,11]. Treatment of these critically ill patients usually brings tremendous pressure to physicians. Pneumonia is the most frequent cause of the above-mentioned septicaemia. Of particular note, the case-fatality rate of nosocomial (i.e., acquired at hospital settings) pneumonia (i.e., hospital-acquired pneumonia (HAP)) is in reality obviously higher than that of community-acquired pneumonia (CAP) [12].

According to the interval between admission and onset of pneumonia, HAP is defined as an infection of the pulmonary parenchyma in patients who acquire the condition at least 48 h after admission to the hospital, or within 14 days after discharge from hospital. The clinical condition of HAP mainly includes the presence of “new lung infiltrate plus clinical evidence that the infiltrate is of an infectious origin, and new-onset fever, purulent sputum, leukocytosis, as well as decline in oxygenation” [11]. By contrast, ventilator-associated pneumonia (VAP) is defined as an infection of pulmonary parenchyma occurring at least 48 h after endotracheal intubation, and also includes the above clinical scenario of HAP. In spite of remarkable advances in the understanding of the contributing causes and prevention, HAP and VAP continue to be frequent complications of hospitalised patients [11]. Additionally, although HAP is considered to be less severe than VAP, serious complications (empyema, septic shock, and multiorgan failure) are observed in approximately 50% of HAP patients, especially those hospitalised in the intensive care unit (ICU) [11]. The incidence rate of VAP amongst mechanically ventilated patients was estimated to range from 9 to 27% [10]. As the microbial resistance loading in hospital settings worsens, the occurrence of VAP may result in significant consequences in association with increasing healthcare consumption and case-fatality rates (the crude mortality rate ranges from 20 to 50%) [10,11,13].

Of the implicated microorganisms causing HAP and VAP, the top five pathogens were *Staphylococcus aureus* (especially, methicillin-resistant *S. aureus* (MRSA)), *Pseudomonas* species (especially *Pseudomonas aeruginosa*), *Acinetobacter* species, *Escherichia coli*, and *Klebsiella* species (including extended-spectrum β-lactamase (ESBL)-producing, and extensively drug-resistant (XDR) Enterobacteriaceae). These pathogens accounted for nearly 80% of all episodes [9,10]. Within the past five years, other less commonly seen but not ignorable pathogens, including (by descending frequency) *Stenotrophomonas maltophilia*, isolates of *Chryseobacterium* species, and *Elizabethkingia meningoseptica*, have also gradually emerged as important HAP/VAP aetiologies [10,11,14,15,16]. *S. aureus* is still ranked as the number one causative pathogen in the US (27.5–36.3%), many states in the European Union (23%) [17,18,19], South Korea and Singapore [11,20]. By stark contrast, in China, Thailand, and Taiwan, Gram-negative bacteria (GNB), including *Pseudomonas aeruginosa*, multidrug-resistant (MDR, defined as a microorganism showing resistance in vitro to three or more antimicrobial classes routinely tested [21]) *Klebsiella pneumoniae*, and MDR-*Acinetobacter baumannii* complex, have all exceeded the prevalence of *S. aureus* as the most commonly causative pathogens of HAP/VAP [10,20].

## 2. Materials and Methods

We first reviewed ample PubMed literature (see references below) documenting the risk factors involved in the acquisition of specific, as well as overall drug-resistant pathogens. In addition, apart from diverse parameters of killing or inhibiting bacterial organisms amongst different antibiotic agents, the antibiotic concentrations at lung parenchymal tissues also have significant impact on their efficacy against pneumonic pathogens. Consequently, the pharmacokinetic (PK) as well as pharmacodynamic (PD) profiles, and clinical efficacy of individual antibiotic agents possibly related to HAP/VAP treatment were also investigated to make appropriate recommendations. Moreover, we also explored the antibiotic combination regimens potentially effective against some notable clinically drug-resistant GNB implicated in HAP/VAP.

## 3. Results

### 3.1. Risk Factors in Association with Acquiring Specific HAP/VAP Pathogens

To optimise antibiotic prescriptions, physicians are obliged to have a thorough understanding of the risk factors associated with the acquisition of respective HA pathogens in the era of high resistance burden. The risk factors regarding important aetiologies of HAP/VAP and adequate antibiotic choice against the most important pathogens, including *S. maltophilia*, *Chryseobacterium indologenes*, and *E. meningoseptica*, are as follows:

#### 3.1.1. MRSA

According to the guidelines recommended by the Infectious Diseases Society of America (IDSA) and the American Thoracic Society (ATS) in 2016, a prior receipt of intravenous (IV) antibiotic agent(s) within 90 days is definitively a risk factor about acquiring MRSA [11]. In addition, if patients are being treated in units where >20% of *S. aureus* isolates showed a methicillin-resistant phenotype, or patients are in units where the MRSA prevalence is unknown, the patients hospitalised at such units were also considered to have a high risk of acquiring MRSA [11]. Another investigation revealed that a higher clinical severity (score of Acute Physiologic and Chronic Health Evaluation (APACHE) II ≥ 12 points) and previous receipt of any antibiotic or surgery [22] were also independent predictors for acquiring VAP due to MRSA. Furthermore, a large-scale European survey observed that patients with a nasopharyngeal colonisation of MRSA were also at risk for subsequently acquiring MRSA infections [23].

#### 3.1.2. *P. aeruginosa*

*P. aeruginosa* is a highly prevalent causative pathogen in HAP/VAP worldwide. This pathogen has a worsening global trend towards more likely displaying MDR phenotypes than before [24]. Inadequate initial treatment of HAP due to *P. aeruginosa* may be associated with increased mortality rates [11]. The presence of an MDR phenotype has been further identified as an independent predictor of an inappropriate initial antibiotic therapy for patients with HAP due to *P. aeruginosa* [25]. The 2016 IDSA/ATS HAP/VAP guidelines also considered the following as important risk factors acquiring MDR-*P. aeruginosa*: A recent receipt of IV antibiotic agent(s) (within 90 days), patients who have chronic obstructive pulmonary disease (COPD), cystic fibrosis, or bronchiectasis, or patients who require ventilator support [8,11]. A Taiwanese study pointed out that patients with bacteraemia due to carbapenem only-resistant *P. aeruginosa* (usually co-exhibiting MDR phenotypes, with a 50% case-fatality rate) mostly had long hospital stay durations (mean, 42.8 days) that were likely related to prior carbapenem use [26]. In addition, a prior fluoroquinolone use was independently associated with subsequent emergence of carbapenem-resistant *P. aeruginosa* in patients (adjusted odds ratio (OR), 4.64; 95% confidence interval (CI), 1.64–13.14; *p* = 0.004) [27]. An investigation revealed that the presence of chronic liver disease was also an independent predictor with respect to acquiring ICU pneumonia due to MDR-*P. aeruginosa* (adjusted OR, 5.43; 95% CI, 1.41–20.89; *p* = 0.014) [28].

#### 3.1.3. XDR-*Acinetobacter* Species

The earliest nosocomial pneumonia caused by carbapenem-resistant *A. baumannii* was reported in Spain in 1998 [29]. Since then, the trend towards an escalating antimicrobial resistance rate amongst isolates of *A. baumannii* complex has become a great concern in clinical settings (especially ICU) in some Asian countries over the last decade [10,20,30,31,32]. By stark contrast, isolates of *A. baumannii* occupied only 2.8% of the implicated organisms among HAP episodes acquired in ICUs in US hospitals between 2015 and 2017 [18]. Apart from the un-interrupted clonal disseminations at some departments of many hospitals and nursing homes in Taiwan [33], the plasmids on *Acinetobacter* species and some clinical *A. baumannii* isolates that harbour genetic determinants encoding the various carbapenem-hydrolysing class D β-lactamases (*bla*_OXA-23_, *bla*_OXA-58_, *bla*_OXA-58_-like, and *bla*_OXA-72_, etc) confer high-level resistance to all carbapenem agents in China, as well as Taiwan [30,34]. Many acquired insertion sequences or transposons have been shown to promote the over-expression, as well as spread of plasmid-associated *bla*_OXA-58_ genes in *Acinetobacter* species [30,34,35]. In addition, a higher point (≥4) of the Charlson co-morbidity index [34,35], a prolonged hospital stay (≥14 days) or an ICU stay (≥10 days) [36,37], a higher APACHE II score (≥16) [37] or Simplified Acute Physiology Score II [38], a recent receipt of broad-spectrum anti-bacterial agents, such as piperacillin-tazobactam, cefepime [39] or any carbapenem agent [37], were all reported to be associated with a predisposition of hospitalised patients to acquire XDR-*A. baumannii* complex pneumonia.

#### 3.1.4. ESBL- and XDR-Enterobacteriaceae Species

Bloodstream infections caused by ESBL-producing Enterobacteriaceae species (especially *Klebsiella pneumoniae*) have been shown to result in higher case-fatality rates in patients than those due to the non-ESBL producers [40]. Of note, Asia has become an epicentre of high ESBL prevalence rates amongst the important Enterobacteriaceae species (mainly *E. coli*, *K. pneumoniae*, and *K. oxytoca*) predominantly implicated in complicated intra-abdominal infections since 2012 [41]. Although Enterobacteriaceae species mainly inhabit the abdominal cavity, they also survive in the oropharynx and respiratory tract as well.

Previous studies suggest that a preceding use of the third-generation cephalosporin agents clearly predisposed hospitalised patients to acquire the infections related to ESBL-producing Enterobacteriaceae species or carbapenem-resistant (CR) Enterobacteriaceae (CRE) [34,42]. In addition, a preceding colonisation of *K. pneumoniae* and *Enterobacter* species in the airway is an independent predictor about developing pneumonia caused by the ESBL-producing Enterobacteriaceae species (OR, 10.96; 95% CI, 2.93–41.0) [43]. Over the last decade, the other notable issue was the emergence of CRE [44]. Between 2015 and 2017, the prevalence of CRE isolates that resulted in HAP ranged from 2.0–3.8% in US hospitals [18,19]. Although hyper-production of ESBL or AmpC β-lactamases plus OmpK36 porin dysfunction is also the common resistance mechanism of some CRE strains [29], the carbapenemase producers amongst CRE are a truly worrisome concern because of their high spread potential. *Klebsiella pneumoniae* carbapenemase (KPC, Ambler class A, especially sequence type (ST) 258 in the US, South America, Greece, Italy, and Israel; and ST11 in Taiwan as well as China) are the most abundant carbapenemases produced by Enterobacteriaceae isolates globally [33,44,45]. The isolates of KPC-2-producing Enterobacteriaceae mostly display high-level resistance to all carbapenem agents [34]. According to prior investigations, severely ill patients who are hospitalised in the ICU [45], and patients who received immunosuppressive agents [41], fluoroquinolone, or extended-spectrum β-lactam agents other than carbapenem [44] were at remarkably high risk for acquiring infections related to KPC-producing *K. pneumoniae*. Furthermore, the use of a nasogastric tube, central venous catheter, urinary catheter or tracheostomy, and a need for mechanical ventilation [46] were also identified as risk factors in association with acquiring CRE septicaemia.

Apart from KPC enzymes, the other carbapenemases that also should be watched are New Delhi metallo-β-lactamases (NDM [34,47], Ambler class B, especially NDM-1) that were originated in the Indian subcontinent. The *bla*_NDM-1_ determinants assisted by unique insertion elements (rather than integrons) in conjunction with a transpose gene between different Enterobacteriaceae species (*K. pneumoniae*, *E. coli*) in the same environment was shown to be responsible for either the clonal spread or in vivo dissemination of strains containing NDM [34]. Although no other risk factors except probable contact with relevant patients were ever reported regarding the acquisition of NDM-producing Enterobacteriaceae strains [48], a cohort contact isolation precaution is currently the best way to prevent the dissemination of *bla*_NDM_-harbouring Enterobacteriaceae isolates.

#### 3.1.5. *S. maltophilia*

*S. maltophilia* is capable of surviving on moist surfaces and producing biofilms, thus it is very difficult to treat. The main MDR mechanisms of *S. maltophilia* include the development of β-lactamase(s), aminoglycoside-modifying enzymes, efflux pumps, and mobile genes exhibiting resistance to trimethoprim-sulfamethoxazole (TMP-SMX) on integrons or plasmids [49]. It has been listed as one of the important MDR pathogens in major hospitals by the World Health Organisation. In the US, *S. maltophilia* accounted for 5.1–6.8% of the aetiologies of overall HAP episodes from 2015 through 2017 [18,19]. For *S. maltophilia* pneumonia, an age ≥65 years-old and receipt of therapy with inappropriate antibiotics were identified as risk factors associated with 30-day mortality amongst patients with cancer [50]. In addition, a Taiwanese study investigating 406 patients with pneumonia caused by *S. maltophilia* observed that only approximately 60% of the enrolled patients ever stayed in the ICU or had a ≥28-day hospital stay before the onset of *S. maltophilia* pneumonia [50]. Moreover, in that Taiwanese study, about one half of these patients received mechanical ventilator support, whilst one quarter of the patients with *S. maltophilia* pneumonia had malignancy, diabetes mellitus, or chronic respiratory disorders [51].

#### 3.1.6. *Chryseobacterium* Species and *E. meningoseptica*

The *Chryseobacterium* species and *E. meningoseptica* are frequently isolated not only from soil, saltwater, and freshwater, but from dry as well as moist surfaces of the clinical environment and equipment [52]. Because of good adaptation ability to the hostile environments, they greatly contribute to extensive contamination of healthcare settings [52]. Although less prevalent than the above HAP pathogens, they have become notable causative organisms amongst diverse septicaemic entities, including pneumonia, bacteraemia, and catheter-associated sepsis in the nosocomial settings [52,53,54].

As seen in other countries [16,52,54,55], *C. indologenes* as well as *E. meningoseptica* have been ranked amongst the top ten causative microorganisms of nosocomial pneumonia for patients hospitalised in the ICU of Taiwanese medical centres. Similar to *C. indologenes*, previous studies investigating resistance mechanisms of *E. meningoseptica* isolates to many β-lactam drugs confirmed that most of them harbour diverse β-lactamase-encoding alleles (*bla*B, *bla*_GOB_, etc.). Unsurprisingly, they also emerged in patients who were hospitalised after recent chemotherapy [53], or were prescribed with a prolonged course(s) of broad-spectrum antibiotic agents (extended-spectrum cephalosporins or carbapenem agents), aminoglycosides, or colistin in clinical settings [16,52,55]. In addition, the other documented risk factors regarding acquisition of pneumonia caused by these pathogens include utilisation of invasive catheters (such as intravascular catheters, or indwelling central venous lines) or non-invasive medical equipment (such as humidifiers) [52,55], underlying co-morbidities of malignancy and/or diabetes mellitus in adults [53,54], and other immunosuppressive conditions or neutropenia regardless of ages [55,56].

With respect to *S. maltophilia*, it is usually susceptible in vitro to TMP-SMX, levofloxacin, moxifloxacin, and tigecycline [49]. These antibiotics are thus good options in the treatment of *S. maltophilia*-related HAP/VAP. By contrast, piperacillin-tazobactam, tigecycline, levofloxacin, and other newer fluoroquinolones (moxifloxacin, garenoxacin, and gatifloxacin) are a good treatment choice against clinical infections due to these two pathogens, whilst TMP-SMX and ciprofloxacin show variable in vitro susceptibility to *C. indologenes* [16,52]. Of interest, vancomycin usually shows good in vitro activity against isolates of *E. meningoseptica*, but when vancomycin is considered in the treatment of severe *E. meningoseptica* infections, a combination with linezolid, ciprofloxacin, or rifampicin is needed [57].

The risk factors regarding acquisition of the above-mentioned six HAP/VAP implicated pathogens are summarised in Table 1. In our opinion, with the exception of cautiously inspecting the administered antibiotic regimens, it is difficult to predict the acquisitions or infections due to ESBL-producing Enterobacteriaceae spp. or CRE isolates by risk factors in advance.

### 3.2. Risk Factors Related to Acquisition of the Overall MDR Pathogens of HAP/VAP

Apart from risk factors of respective HAP/VAP aetiologies, when empirical antibiotic choice is prescribed for patients with HAP or VAP, the clinical severity as well as the epidemiology, and the microbial resistance loading of given healthcare settings need to be cautiously evaluated [10,11]. For patients with critically ill conditions (haemodynamically unstable status, high short-term mortality rates) or at high risk of acquiring MDR pathogens, prescription of a combination regimen against the troublesome bacteria has apparently been accepted worldwide [10,11,14,34,58,59,60,61]. The risk factors for patients likely acquiring VAP due to overall MDR pathogens, as specifically stressed by the 2016 IDSA/ATS HAP/VAP guidelines, are as follows: (1) Receipt of IV antibiotic within 90 days, (2) complicated septic shock, (3) acute respiratory distress syndrome preceding VAP, (4) acute renal replacement therapy prior to VAP onset, and (5) five or more days of hospitalisation before the occurrence of VAP [11]. An antibiotic treatment for MRSA (glycopeptide or linezolid) should be considered to be added if patients are colonised with this pathogen within the airway, or are admitted to a unit where a high (≥20%) prevalence of MRSA amongst *S. aureus* isolates is detected [10,11,14].

### 3.3. Special Considerations about Specific Antibiotics for HAP/VAP

Doripenem has an in vitro spectrum similar to meropenem [62]. However, it is noteworthy that a 1 g doripenem administered for a 4-h infusion duration every 8 h for 7 days was not shown to have definitive clinical superiority to the 10-day imipenem/cilastatin regimen prescribed with a dosage of 1000 mg every 6 h in treating late-onset VAP, in terms of all-cause 28-day mortality (21.5% vs. 14.8%; 95% CI, −5.0 to 18.5%) and the Kaplan–Meier survival curve analysis for VAP caused by *P. aeruginosa* (*p* = 0.040) [63]. In addition, for patients with haematological diseases with HAP or VAP, a doripenem dosage of 500 mg intravenously every 8 h was not shown to have a better survival rate compared to meropenem (1000 mg every 8 h) [64]. The US Food and Drug Administration (FDA) and the European Medicines Agency (EMA) do not presently recommend doripenem for the treatment of HAP and VAP [11,14]. Ceftobiprole, a first anti-MRSA cephalosporin having a spectrum covering many HAP pathogens, was shown to have higher mortality rates compared to ceftazidime amongst VAP patients [11]. Additionally, ceftaroline shows excellent in vitro data and clinical efficacy against CA-MRSA isolates and was approved in treatment of CAP in 2010 [65]. Nevertheless, ceftaroline is not recommended in the management of HA-MRSA due to the high minimum inhibitory concentration (MIC; 8 mg/L) [66]. As compared to vancomycin, telavancin was also shown to have higher case-fatality rates amongst MRSA-VAP patients who had significant renal dysfunction (i.e., creatinine clearance < 30 mL/min) [11]. There are no studies to elucidate the efficacy of tedizolid against MRSA HAP [11]. The above-mentioned drugs are not recommended in the treatment of MRSA-HAP/VAP presently. The recommended antibiotic regimens and dosage (calculated by creatinine clearance rates ≥ 50 mL/min) against HAP/VAP with suspicious relevance to *P. aeruginosa* are presented in Table 2.

As issued by the 2016 IDSA/ATS HAP/VAP guidelines [11], the PK and PD data are also important considerations for prescribing effective antibiotics in the treatment of MDR or XDR respiratory GNB pathogens. Herein, we summarise the PK/PD data of important antibiotics of last resort.

In spite of a bacteriostatic nature and a relatively low serum concentration under standard-dose administration (100 mg loading followed by 50 mg every 12 h), tigecycline is active in vitro against most CRE and some CR-*A. baumannii* complex isolates [68], and is thus frequently adopted as an adjuvant (combined with meropenem and colistin) in treating many nosocomial XDR-GNB, especially producers of KPC [69]. Of note, treatment with tigecycline at a dosage of 200 mg loading followed by 100 mg every 12 h showed numerically better cure rates than imipenem/cilastatin (1 g every 8 h) amongst patients with HAP. This result was attributed to a higher ratio of area under the concentration–time curve over 24 h (AUC_0–24_) divided by the MIC (AUC_0–24_/MIC) than that of the subgroup treated with the standard-dose regimen [70], and a significantly higher penetration into infected lung parenchyma [71]. Furthermore, the high-dose regimen of tigecycline was also shown to provide a significant benefit of survival and have appropriate safety in treatment of the non-haemodialysis patients with CRE septicaemia [72]. Despite a better PK parameter achieved by the high-dose tigecycline, approximately one quarter (23.5%) of patients who received a ≥7-day tigecycline monotherapy were reported to have superinfections due to *P. aeruginosa* [73]. Consequently, to avoid the late-onset pseudomonal superinfections, a combination of one anti-pseudomonal antibiotic with tigecycline is clinically meaningful when tigecycline is administered.

The revival of polymyxin B and colistin (polymyxin E) provide another choice in the treatment against important XDR-GNB infections since the last decade. The optimum dosage of IV colistin was proposed to maximise its efficacy against the infections related to MDR-GNB (with colistin MIC ≥ 1 mg/L) [67]. Additionally, during the interval of an aerosolised colistimethate sodium (CMS) dosing (2 million units [MU]), a high pulmonary AUC of colistin (ranging 18.9–73.1 μg h/mL) and a high maximum pulmonary colistin concentration (6.00 ± 3.45 μg/mL) are achieved in humans [74]. Nevertheless, there are divergent recommendations about dosage of IV drip and inhaled colistin in treatment of various GNB infections [67,69,74,75,76,77]. For example, an adjuvant 1 MU aerosolised CMS administered via either jet or ultrasonic nebuliser every 8 h in conjunction with IV colistin therapy was observed to improve the clinical cure rates of VAP caused by colistin-only susceptible GNB [73]. By contrast, despite existence of controversy and the technical requirements of using CMS [78,79], a high-dose regimen of aerosolised CMS (4 MU administered every 8 h) monotherapy also showed benefits in improving clinical parameters (in terms of improvement in pulmonary oxygenation function, and shorter durations of ventilator use), as well as earlier GNB eradications, whilst there was no increase in acute nephrotoxicity compared to IV CMS (4.5 MU every 12 h following 9 MU loading) in treating VAP caused by MDR-GNB (*P. aeruginosa* and *A. baumannii* complex spp. predominantly) [77]. In the present era of the antibiotic pipeline, however, this drug is better reserved for treatment against CRE or CR-*A. baumannii* infections, or for HAP patients who are at high risk of acquiring MDR organisms [11,14,33,80]. Compared to polymyxin E, a significantly higher proportion of CMS is converted into colistin in vivo for polymyxin B [81]. Nevertheless, polymyxin B is not recommended as a treatment choice against GNB-related HAP/VAP presently by any expert worldwide.

Although formally approved as a single drug treatment of only urinary tract infections (UTI) caused by MDR-GNB [82], fosfomycin also has a wide in vitro anti-bacterial spectrum and was shown to have an excellent percentage of penetration into infected lung tissue [83,84]. Consequently, herein, we suggest fosfomycin as a potentially adjunctive option in combination with other antibiotics in the treatment of HAP/VAP due to CRE and MDR-*P. aeruginosa*, too.

In this decade, ceftazidime-avibactam, a new drug having a spectrum superior to all carbapenems against XDR-GNB, was approved by the US FDA and the EMA to treat HAP/VAP related to many CRE (producers of KPC and/or oxacillinase [OXA]-48, 181-like enzymes) and CR-*P. aeruginosa* [85,86,87]. Thus, it is suitable to be regarded as the first-line antibiotic when infections caused by the ESBL producers and/or CRE are highly suspected. Although ceftazidime-avibactam is effective in vitro against isolates of ESBL, KPC, AmpC β-lactamase, and OXA-48-like producers of Enterobacteriaceae, it lacks the activity against isolates of Gram-positive bacteria, metallo-β-lactamase (MβL) producers of Enterobacteriaceae spp., and XDR-*A. baumannii* complex spp. [85]. A combination of other antibiotic agents (e.g., plazomicin, fosfomycin, etc.) [88] or the anti-MRSA agent with ceftazidime-avibactam is plausibly recommended to enforce control of drug-resistant VAP.

The other novel antibiotic ceftolozane-tazobactam was shown to have an excellent in vitro activity against XDR-*P. aeruginosa* isolates [89] and a good PK profile in the thorax [90], but less in vitro activity against ESBL (especially SHV type)-producing *K. pneumoniae* than most carbapenems [89]. Apart from being effective against the organisms causing complicated intra-abdominal infections and UTI, it was also shown to have adequate clinical efficacy in treating HAP/VAP [19]. In June 2019, this drug was also approved for HAP/VAP therapy by the US FDA.

Cefiderocol, a siderophore (catechol moiety, chelating ferric ions)-containing cephalosporin agent modified from ceftazidime, has a good PK profile (except in the abdominal cavity) in humans, and thus is a promising antibiotic in treatment of HAP/VAP caused by many important XDR-GNB, including KPC and most MβL producers, CR-*P. aeruginosa* and CR-*A. baumannii* isolates [85,91]. This novel agent was approved by the US FDA in the treatment of UTIs caused by XDR-GNB in November 2019, but it is not yet approved for HAP/VAP treatment.

Although aerosolised amikacin initially showed a fantastic in vitro effect on many MDR-GNB [92], its clinical trial was terminated due to poor clinical efficacy. Additionally, systemic amikacin administration (15–20 mg/kg once daily) was not able to achieve acceptable PK and PD parameters regarding the penetration into epithelial lining fluid and killing of pulmonary GNB [93]. In our opinion, as nephrotoxicity and the poor PK data in lung tissue likely outweigh its therapeutic benefits, prescription of IV drip amikacin is not strongly recommended against HAP/VAP unless there is an emergence of complicated bacteraemia or a concomitant UTI due to MDR-GNB showing an in vitro susceptibility to amikacin.

The regimens and dosage of antibiotics recommended against HAP/VAP related to XDR-*A. baumannii* complex and CRE are illustrated in Table 3.

### 3.4. Optimal Treatment Durations, Including the Combination Antibiotic Regimens

Two notable differences were observed between the HAP/VAP guidelines recommended by the IDSA/ATS in 2016 and Europe/Latin America in 2017 [11,14]. Firstly, to alleviate the resistance burden, the IDSA/ATS guidelines considered a <7-day antibiotic therapy is the most favourable duration, but its accurate duration might be dependent upon the patient’s clinical response, and improvement of radiological and laboratory parameters [11]. By contrast, the European guidelines advocated that longer (>7 days) courses of antibiotic(s) were needed in patients with immunodeficiency, structural lung defects, necrotising pneumonia, empyema, inappropriate initial empiric therapy, or had XDR-GNB infections or bacteraemia [14]. Secondly, prescription of the regimen of dual anti-GNB antibiotics against HAP/VAP caused by the MDR-GNB was favoured by the American guidelines for patients who have septic shock or in-hospital mortality rates (>25%). However, partly corresponding to prior studies with respect to relapsed septicaemia [95,96], the Europe/Latin America guidelines suggested that prolonged therapy durations (>7 days) of the regimen of dual anti-GNB antibiotic combination regimen were needed to treat HAP/VAP caused by MDR, XDR, or pandrug-resistant GNB (such as CRE, or glucose non-fermenting isolates, e.g., XDR-*A. baumannii* complex species and CR-*P. aeruginosa*), to treat patients who were immunosuppressed hosts (with neutropenia or recipients of stem cell transplantation), patients who have persistently unstable haemodynamic conditions, patients who received initial inappropriate antibiotic therapy, or required use of the second-line broad-spectrum antibiotic agents (e.g., tigecycline, colistin, etc.) [14].

The possible causes of failure of responding to initial antibiotic therapy against HAP/VAP are illustrated in Table 4. In addition, we provide an algorithm of the strategies about antibiotic therapy for HAP/VAP, as illustrated in Figure 1.

## 4. Discussion

In addition to considerations of efficacy of specific antibiotics, only a few randomised control studies were published on recommendations for antibiotic treatment against HAP/VAP caused by MDR-GNB [14,34,85]. A Spanish study investigating the outcomes of ICU patients with VAP caused by *P. aeruginosa* observed that initial use of antibiotic combination therapy (mainly anti-pseudomonal β-lactam plus an aminoglycoside or an anti-pseudomonal fluoroquinolone agent) indeed significantly reduced the likelihood of inappropriate therapy, which was strongly associated with higher case-fatality risk [97]. Nevertheless, in distinction from two guidelines [11,14], there were many diversified regimens of antibiotic combinations proposed against clinical infections due to isolates of CR-*Acinetobacter baumannii* complex species [59,75,80,98,99,100], as well as CRE [34,69,94,101]. Despite no reduction of mortality rates [11], a useful de-escalation of an antibiotic has the benefit of decreasing resistance burden. It should be seriously considered for patients who have definitively positive culture data and had an improvement in clinical condition. Multiplex polymerase chain reaction tests are beneficial in assistance of accurately diagnosing the implicated aetiologies of pneumonia in ICU [84,102].

Apart from adequate antimicrobial treatment, of paramount importance is a strict implementation of antibiotic stewardship policy, including education to primary-care staff on preventing the dissemination of high-risk hospital microorganisms, such as cohort isolation, and optimisation of the appropriate doses of antibiotics, etc. [11,14,85,103]; as well as an in-time de-escalation of antibiotic therapy according to culture data, once HAP/VAP has shown significant improvement after adequate therapy [11,14]. Additionally, to quickly detect the existence of carbapenemase-encoding alleles in XDR-GNB isolates, polymerase chain reaction tests provide great help in achieving the goal of “precision medicine” that avoids the erroneous prescription of novel antibiotics and elucidates the resistance epidemiology (especially for CRE isolates) [85,104].

In addition to effective antibiotic therapy, there are many other important measures with regard to preventing the development of HAP/VAP for hospitalised patients. A well-designed study showed that, if there were no contraindications, cautious adoptions of the mode of non-invasive positive pressure ventilation amongst patients with COPD or cardiogenic pulmonary oedema was beneficial in decreasing the incidence and mortality rates of HAP [105]. In addition, in order to reduce lengths of ICU stay and improve mortality rates related to VAP, the US Institute for Healthcare Improvement grouped the interventions together in 2012 for the consistent prevention of VAP (i.e., VAP care bundle) [10,13]. In brief, to ensure patient comfort, the prescription of dexmedetomidine or propofol as a sedative choice is preferred to benzodiazepine, as well as a neuromuscular blocking agent when possible, for patients who are mechanically ventilated [13]. Moreover, the decreased maintenance dose of IV sedative drugs when possible, implementation of protocol-guided daily sedation interruption and spontaneous breathing trials to assess the readiness of extubation, and early physical as well as occupational therapy initiated by a mobility team were shown to effectively shorten the duration of ventilator use (2.4–3.1 days) and length of ICU stay (3.5–3.8 days) [13,106,107,108,109,110].

Some of the mechanically ventilated patients have obtunded consciousness. Consequently, the sources of potential microorganism inoculation for VAP likely originate in the sinuses, oropharynx, subglottic area, and the upper gastrointestinal tract to a substantial degree [13]. Consequently, the use of subglottic secretion drainage (SSD) for the patients expected to be intubated for >48 h, maintenance of the target pressure of the endotracheal cuff at about 25 cmH_2_O (to prevent microaspiration of gastric contents), and maintenance of the semi-recumbent position (i.e., elevation of the head above the bed up to 30–45 degrees) when possible were also significantly beneficial for patients receiving mechanical ventilation in terms of decreasing the VAP incidence rate and length of ICU stay [10,13,111,112,113]. To reduce channel formation and fluid leakage from the subglottic area, the use of an endotracheal tube with an ultrathin polyurethane cuff in conjunction with SSD was demonstrated to help prevent early- and late-onset VAP to significant degrees [114]. Furthermore, an application of antiseptic chlorhexidine gluconate once daily to selectively decontaminate the oral cavity for patients undergoing mechanical ventilation has been gradually accepted as an effective method in reducing the VAP incidence rate (44%) [13,115]. Of interest, the other alternative strategy also employed amongst the mechanically ventilated patients to broadly decontaminate the oropharyngeal and digestive tract is administration of prophylactic non-absorbable probiotics (containing *Pediococcus pentosaceus*, *Leuconostoc mesenteroides*, and *Lactobacillus* spp., etc). It was shown by convincing evidence to lead to a significant reduction in the incidence of VAP, *Clostridium difficile*-associated diarrhoea, and length of ICU stay as well, with no significant difference in hospital mortality [116,117]. Nevertheless, the worrisome concern of increasing MDR bacterial loading in the ICU after widespread utilisation of probiotics [14] remains to be alerted. Finally, cautious prescription of proton-pump inhibitors (e.g., pantoprazole) should be exercised amongst ICU patients, as they have a three-fold increased risk of developing VAP compared to those receiving histamine H2 receptor antagonists [118].

Apart from correct prescription of antibiotics for effective treatment of nosocomial pneumonia, the measures beneficial in preventing the development of HAP/VAP are summarised in Table 5.

## 5. Conclusions

To determine whether to use an antibiotic combination or not, we need to first cautiously evaluate the risks of acquiring MDR pathogens and clinical severity first when facing patients with HAP or VAP. A few pneumonic MDR or XDR-GNB pathogens do warrant >7-day durations of antibiotic(s) therapy. Although novel anti-XDR-GNB antibiotics have been launched recently, the antibiotics of last resort are valuable and should be prescribed with more discretion in the present MDR era. In addition, an antibiotic stewardship in combination with other recommended measures needs to be strictly implemented to decrease the incidence rate of HAP/VAP.

## Figures and Tables

**Figure 1 jcm-09-00275-f001:**
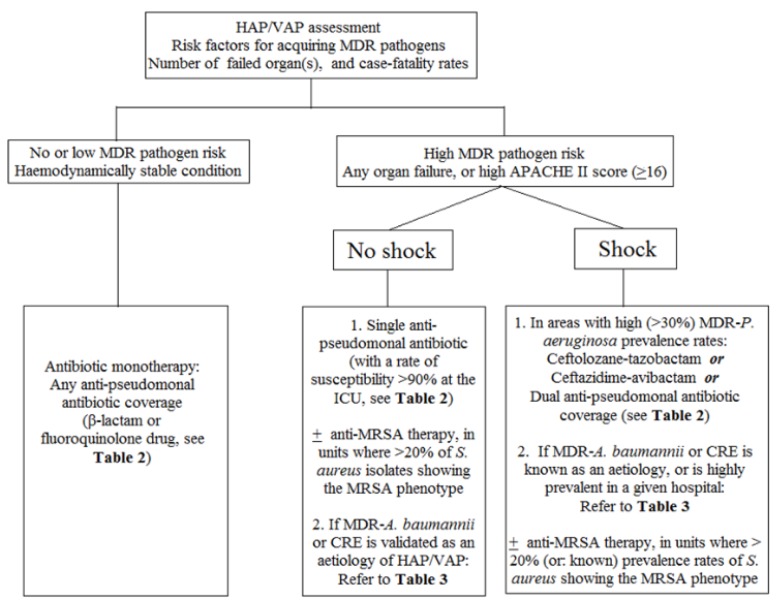
Strategies of antibiotic therapy for hospital-acquired pneumonia or ventilator-associated pneumonia. HAP, hospital-acquired pneumonia. VAP, ventilator-associated pneumonia. MDR, multidrug-resistant. APACHE, Acute Physiologic and Chronic Health Evaluation. ICU, intensive care unit. MRSA, methicillin-resistant *Staphylococcus aureus*. CRE, carbapenem-resistant Enterobacteriaceae.

**Table 1 jcm-09-00275-t001:** The risk factors regarding acquisition of the following clinical multidrug-resistant bacteria.

MDR Bacteria	Risk Factors
MRSA	Stay at a given unit where there is a >20% prevalence of methicillin resistance amongst clinical *S. aureus* isolates [11,14]
	A receipt of intravenous antibiotic(s) within 90 days [11,22]
	Higher clinical severity (APACHE II score), or prior receipt of surgery [22]
	Delay-onset pneumonia at hospital, a nasopharyngeal colonisation of MRSA [23]
MDR- or CR-*Pseudomonas aeruginosa*	More than 10% prevalence of resistance to a single anti-pseudomonal antibiotic class amongst clinical *P. aeruginosa* isolates at a specific unit [11]
	Receipt of intravenous antibiotic(s), especially carbapenem or fluoroquinolone agents within 90 days [11,26,27] Prolonged (>3 weeks) hospital stay durations before HAP [26]
	The presence of chronic hepatic disorder, diabetes mellitus, or admission to intensive care units [28]
XDR- or CR-*Acinetobacter baumannii* complex species	Stay at a unit where isolates of XDR-*A. baumannii* complex are highly prevalent [33]
	Charlson co-morbidity index ≥4 points [34,35]
	Prolonged (≥14-day) hospital stays, or ≥10-day ICU stays [36,37]
	A high APACHE II score (≥16) or Simplified Acute Physiology Score II [37,38]
	Prior receipt of cefepime, piperacillin-tazobactam, or carbapenem agents [37,39]
ESBL-producing or carbapenem-resistant Enterobacteriaceae species	Stay at an institute where NDM-producing Enterobacteriaceae are highly prevalent, or contact with patients who are colonised with *bla*_NDM_-harbouring Enterobacteriaceae isolates [34]
	Receipt of immunosuppressive agent(s) [41]
	Prior colonisation of drug-resistant isolates of *K. pneumoniae* or *Enterobacter* species within the airway [43]
	Receipt of fluoroquinolone or extended-spectrum cephalosporins [44]
	High-severity residents requiring hospitalisation at ICUs [45]
*Stenotrophomonas maltophila*	An ICU stay, or >28-day hospital stay course, or required ventilator use, with co-morbidities such as malignancy or diabetes mellitus, etc. [51]
*Chryseobacterium* species, or *Elizabethkingia meningoseptica*	Recent receipt of extended-spectrum cephalosporin, carbapenem, aminoglycoside, or colistin therapy [16,52,55]
	Use of intravascular catheter or indwelling central venous lines, or other non-invasive equipment (e.g., humidifiers) [52,55]
	Recent receipt of chemotherapy [53]
	Underlying co-morbidities of malignancy, or diabetes mellitus in adults [53,54]
	Immunosuppressed conditions, or neutropenia regardless of ages [55,56]

MDR, multidrug-resistant. MRSA, methicillin-resistant *Staphylococcus aureus*. APACHE, Acute Physiologic and Chronic Health Evaluation. CR, carbapenem-resistant. HAP, hospital-acquired pneumonia. XDR, extensively drug-resistant. ICU, intensive care unit. ESBL, extended-spectrum β-lactamase. NDM, New Delhi metallo-β-lactamase.

**Table 2 jcm-09-00275-t002:** The recommended regimens and dosage (if creatinine clearance rates ≥ 50 mL/min) for patients with hospital-acquired pneumonia of which aetiologies are likely related to *Pseudomonas aeruginosa* and/or methicillin-resistant *Staphylococcus aureus.*

Clinical Severity and Risk Evaluation	Recommended Antibiotic(s)
Haemodynamically stable, low MDR-GNB risks	Any anti-pseudomonal agent (except aminoglycoside IVD monotherapy)
Haemodynamically not stable, or higher risks of acquiring MDR-GNB pathogens	**Monotherapy with any of the following antibiotics, including:**
	Ceftolozane-tazobactam: 1.5 g IVD every 8 h [19]
	Ceftazidime-avibactam: 2.5 g IVD every 8 h Or
	Piperacillin-tazobactam: 4.5 g IVD (EI) every 6 h
	Ceftazidime: 2 g IVD (EI) every 8 h
	Cefepime: 2 g IVD (EI) every 12 h or every 8 h
	Imipenem/cilastatin sodium: 500 mg IVD every 6 h or 1 g IVD every 8 h
	Meropenem: 1–2 g IVD (EI) every 8 h
	Cefoperazone-sulbactam: 4 g IVD every 12 h
	+(any of the below non-β-lactam agent) Ciprofloxacin: 400 mg IVD every 8 h (preferred), or alternatively levofloxacin: 750 mg once daily
	Colistin (66.8 mg/vial): 5 mg/kg IVD loading, then 2.5 mg × (1.5 × CrCl + 30) IVD every 12 h [67] ±Aerosolised colistimethate sodium (2 MU/vial): 1–2 vials every 12 h or 8 h, or ±Amikacin: 15–20 mg/kg IVD once daily, if complicated bacteraemia, combined with urinary tract infection, and in vitro susceptible to amikacin
High risk of acquiring MRSA pneumonia	Vancomycin: 25–30 mg/kg loading, then 15 mg/kg IVD every 12 h, or
	Teicoplanin: 12 mg/kg every 12 h × 3 doses (loading), then 6–12 mg/kg IVD once daily, or
	Linezolid: 600 mg IVD every 12 h

MDR-GNB, multidrug-resistant Gram-negative bacteria. IVD, intravenous drip. MU, million units. EI, extended infusion (intravenous drip for 3 h). CrCl, creatinine clearance rate. MRSA, methicillin-resistant *Staphylococcus aureus*.

**Table 3 jcm-09-00275-t003:** The recommended antibiotic regimens and dosage (if creatinine clearance rates > 50 mL/min) for patients with hospital-acquired or ventilator-associated pneumonia related to extensively drug-resistant *Acinetobacter baumannii* complex species and carbapenem-resistant Enterobacteriaceae.

Causative Organisms	Recommended Antibiotic(s)
CR- or XDR-*Acinetobacter baumannii* complex	Ampicillin/sulbactam (0.5/1 g/vial): 3 g IVD every 6 h (if in vitro susceptible and haemodynamically stable)
	Aerosolised colististimate sodium (2 MU/vial): 2 vials every 8 h (if in vitro susceptible and haemodynamically stable)
	Otherwise High-dose meropenem [EI], or doripenem [EI], or imipenem/cilastatin, plus sulbactam: 2.0 g IVD every 6 h, or alternatively colistin (66.8 mg/vial): 2.5–5.0 mg/kg/day IVD (divided into 2–3 times per day, if normal renal function) [75]
	±Aerosolised colistimethate sodium (2 MU/vial): 1–2 vials every 12 h or every 8 h, or ±Amikacin: 15–20 mg/kg IVD once daily, if complicated bacteraemia and/or urinary tract infection, and in vitro susceptible to amikacin
	Tigecycline: 50 mg IVD every 12 h (after 150–200 mg loading) plus any anti-pseudomonal carbapenem (EI if necessary)
CR-Enterobacteriaceae spp.	Regardless of haemodynamic condition or severity— Tigecycline: 50 mg IVD every 12 h (after 150–200 mg loading) plus Meropenem: 2 g IVD [EI] every 8 h, and colistin (66.8 mg/vial): 1 vial IVD every 8 h, or 2 vials IVD every 12 h after adequate dose loading if CrCl is normal (or alternatively, Fosfomycin: 2 g IVD every 6 h)
	Ceftazidime-avibactam: 2.5 g IVD every 8 h (against KPC, or partial oxacillinase-producing CRE)
	Dual carbapenem regimen (ertapenem: 1 g IVD once daily plus high-dose meropenem or doripenem [EI]) against KPC producers that are in vitro resistant to colistin [94]

EI, extended infusion (intravenous drip for 3 h). CrCl, creatinine clearance rate. CR, carbapenem-resistant. XDR, extensively drug-resistant. IVD, intravenous drip. MU, million units.

**Table 4 jcm-09-00275-t004:** Possible causes of failure of responding to initial antibiotic therapy against hospital-acquired and/or ventilator-associated pneumonia.

Inadequate spectrum of antimicrobial(s) Inadequate dosage prescription of antibiotic(s) Lack of, or insufficient control of, the source of HAP/VAP (e.g., inadequately drained empyema, extrapulmonary source) Specific individual factors, including: (1) High clinical severity (e.g., high APACHE II point) (2) Immunocompromised condition (3) Inadequate duration of antibiotic therapy (4) Incorrect diagnosis about HAP/VAP (e.g., congestive heart failure, pulmonary embolism)

HAP, hospital-acquired pneumonia. VAP, ventilator-associated pneumonia. APACHE, Acute Physiologic and Chronic Health Evaluation.

**Table 5 jcm-09-00275-t005:** Measures of preventing the development of nosocomial pneumonia apart from correct antibiotic prescription.

Use of non-invasive positive pressure support if feasible. Avoidance of using benzodiazepine and neuromuscular-blocking agent for intubated patients as possible. Decrease maintenance dose of intravenous sedative agents as possible. Implementation of protocol-guided daily sedation interruption, as well as spontaneous breathing trials (to assess the readiness of extubation). Use of subglottic secretion drainage for the patients expected to be intubated for >48 h. Maintenance of the target pressure of the endotracheal cuff at about 25 cm H_2_O. Maintenance of the semi-recumbent position (head above the bed, up to 30–45 degrees). Application of antiseptic chlorhexidine gluconate once daily, to selectively decontaminate the oral cavity for intubated patients. Administration of prophylactic oral non-absorbable probiotics. Cautious prescription of proton-pump inhibitors as possible.

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
