# Peer review of "Epidemiology, Treatment, and Prevention of Nosocomial Bacterial Pneumonia"

_jcm, 2020, doi:10.3390/jcm9010275_

Round 1
Reviewer 1 Report
The manuscript is interesting, approaching current aspects of responsible pathogens, their drugs resistance, and includes less usual, emerging ones.
However, the paper should be improved in some aspects:
I miss a definition of HAP and VAP. It could be the one included in the recent Guidelines of ATS (American Thoracic Society, Infectious Diseases Society of America. Guidelines for the management of adults with hospital-acquired, ventilator-associated, and healthcare-associated pneumonia. Am J Respir Crit Care Med 2005;171(4):388-416. Kalil AC, Metersky ML, Klompas M, Muscedere J, Sweeney DA, Palmer LB, et al. Management of Adults With Hospital-acquired and Ventilator-associated Pneumonia: 2016 Clinical Practice Guidelines by the Infectious Diseases Society of America and the American Thoracic Society. Clin Infect Dis Off Publ Infect Dis Soc Am 2016;63(5):e61-111.): “new lung infiltrate plus clinical evidence that the infiltrate is of an infectious origin, which include the new onset of fever, purulent sputum, leukocytosis, and decline in oxygenation.” The Materials and methods section is too short, and should be amplified. There is a mistake in lines 95-97 (Results section). It can be read: “The risk factors regarding important aetiologies of HAP/VAP and adequate antibiotic choice against S. maltophilia, Chyrseobacterium indologenes and E. meningoseptica are as follows:”. The paragraph could be for instance: “The risk factors regarding important aetiologies of HAP/VAP and adequate antibiotic choice against the most important pathogens, including maltophilia, Chyrseobacterium indologenes and E. meningoseptica are as follows:” Some new antibiotics, as Telavancin or Tedizolid, have not been mentioned, and Ceftobitrole and Ceftaroline deserve a little more information. In general, these and other new antibiotics, like Ceftazidime-Avibactam and Ceftolozane-Tazobactam should be approached comparing them with the more traditional ones used for HAP/VAP. Deescalation of antibiotic therapy is an important issue that must be pointed out more extensively. A table indicating possible causes of failure of initial antibiotic therapy can be included. An algorithm simplifying the strategies of therapy for HAP/VAP, according to risk for MDR pathogens, severity and mortality may also be included. A good example can be found in: Torres A, Niederman MS, Chastre J, et al. International ERS/ESICM/ESCMID/ALAT guidelines for the management of hospital-acquired pneumonia and ventilator-associated pneumonia. Eur Respir J 2017;50(3). An important concern is that, as far as I know, there is not enough evidence to recommend Aerosolised Colistimetate as monotherapy in HAP/VAP, and we need randomized clinical trials to asses its true role in the management of pneumonias related to MDR pathogens (Zampieri FG, Nassar AP, Gusmao-Flores D, Taniguchi LU, Torres A, Ranzani OT. Nebulized antibiotics for ventilator-associated pneumonia: a systematic review and meta-analysis. Crit Care 2015;19:150. Rello J, Rouby JJ, Sole-Lleonart C, Chastre J, Blot S, Luyt CE, et al. Key considerations on nebulization of antimicrobial agents to mechanically ventilated patients. Clin Microbiol Infect 2017;23(9):640-646.). The same thing happens with Fosfomycin. The tables 2 and 3 must be modified, as well as the discussion about these issues. Furthermore, these tables don´t show clearly when we have to choose either a monotherapy or a combined antibiotic treatment.
Author Response
Reviewer 1 -----
The manuscript is interesting, approaching current aspects of responsible pathogens, their drugs resistance, and includes less usual, emerging ones.
However, the paper should be improved in some aspects:
I miss a definition of HAP and VAP. It could be the one included in the recent Guidelines of ATS (American Thoracic Society, Infectious Diseases Society of America. Guidelines for the management of adults with hospital-acquired, ventilator-associated, and healthcare-associated pneumonia. Am J Respir Crit Care Med 2005;171(4):388-416. Kalil AC, Metersky ML, Klompas M, Muscedere J, Sweeney DA, Palmer LB, et al. Management of Adults With Hospital-acquired and Ventilator-associated Pneumonia: 2016 Clinical Practice Guidelines by the Infectious Diseases Society of America and the American Thoracic Society. Clin Infect Dis Off Publ Infect Dis Soc Am 2016;63(5):e61-111.): “new lung infiltrate plus clinical evidence that the infiltrate is of an infectious origin, which include the new onset of fever, purulent sputum, leukocytosis, and decline in oxygenation.”
ANS: We revise it accordingly.
The Materials and methods section is too short, and should be amplified.
ANS: We revise them (add some important information about searching Pubmed data) accordingly.
There is a mistake in lines 95-97 (Results section). It can be read: “The risk factors regarding important aetiologies of HAP/VAP and adequate antibiotic choice against S. maltophilia, Chyrseobacterium indologenes and E. meningoseptica are as follows:”. The paragraph could be for instance: “The risk factors regarding important aetiologies of HAP/VAP and adequate antibiotic choice against the most important pathogens, including maltophilia, Chyrseobacterium indologenes and E. meningoseptica are as follows:”
ANS: We thank the reviewer’s suggestion, and we revise them accordingly.
Some new antibiotics, as Telavancin or Tedizolid, have not been mentioned, and Ceftobitrole and Ceftaroline deserve a little more information.
ANS: We add the relevant information regarding these antibiotics accordingly in this revised manuscript.
In general, these and other new antibiotics, like Ceftazidime-Avibactam and Ceftolozane-Tazobactam should be approached comparing them with the more traditional ones used for HAP/VAP.
ANS: We thank the reviewers’ valuable suggestions, and revise them accordingly in our revised manuscript.
De-escalation of antibiotic therapy is an important issue that must be pointed out more extensively.
ANS: We add the relevant statement in the Discussion section accordingly.
A table indicating possible causes of failure of initial antibiotic therapy can be included.
ANS: We create one table (new Table 4) to illustrate the potential causes of failure of initial antibiotic therapy accordingly.
An algorithm simplifying the strategies of therapy for HAP/VAP, according to risk for MDR pathogens, severity and mortality may also be included. A good example can be found in: Torres A, Niederman MS, Chastre J, et al. International ERS/ESICM/ESCMID/ALAT guidelines for the management of hospital-acquired pneumonia and ventilator-associated pneumonia. Eur Respir J 2017;50(3).
ANS: We add the algorithm (new Figure 1) accordingly in the revised manuscript.
An important concern is that, as far as I know, there is not enough evidence to recommend Aerosolised Colistimetate as monotherapy in HAP/VAP, and we need randomized clinical trials to assess its true role in the management of pneumonias related to MDR pathogens (Zampieri FG, Nassar AP, Gusmao-Flores D, Taniguchi LU, Torres A, Ranzani OT. Nebulized antibiotics for ventilator-associated pneumonia: a systematic review and meta-analysis. Crit Care 2015;19:150. Rello J, Rouby JJ, Sole-Lleonart C, Chastre J, Blot S, Luyt CE, et al. Key considerations on nebulization of antimicrobial agents to mechanically ventilated patients. Clin Microbiol Infect 2017;23(9):640-646.).
ANS: We thank the reviewer’s valuable suggestions, and decide to add the suggested information, and delete its monotherapy recommendation in Table 2 in the revised manuscript.
The same thing happens with Fosfomycin.
ANS: In this revised manuscript, we would like to change “recommend” into “suggest” about prescription of fosfomycin as a combination antibiotic with others (not monotherapy) in the treatment of MDR-GNB-related HAP/VAP because of its good PK profile. In addition, we also preserve its role as conjunctive option against CRE in Table 3.
The Table 2 and 3 must be modified, as well as the discussion about these issues. Furthermore, these tables don´t show clearly when we have to choose either a monotherapy or a combined antibiotic treatment.
ANS: Table 2 has clearly written that if patients have risk about acquiring MDR-P. aeruginosa and the haemodynamically unstable condition, it is warranted to precribe ceftolozane-tazobactam (monotherapy) or any of combined regimens; in addition, given the prevalence of MRSA is high (>20%) at a hospitalization unit, prescription of the antibiotic covering MRSA should be considered. We add some brief statements in Table 3 about the timing of clinical use in this revised manuscript.

Reviewer 2 Report
Jean et al. made a narrative review on the treatment of nosocomial pneumonia.
Although the topic is interesting, it seems to me that the authors should have gone a little bit further into the subject.
Here are a few comments
Title does not reflect the review : I suggest the authors to modify it to Epidemiology, treatment and prevention of nosocomial bacterial pneumonia There is no mention of the burden of respiratory viruses in nosocomial pneumonia “Nosocomial” should be defined “MDR P. aeruginosa” should be defined. I guess table 2 reflect probabilistic treatment ? there is no distinction between ESBL at risk patient and CR enterobacteriae. I do not understand what are the different choices of treatment ? Do the authors propose Ceftolozane-tazobactam and Ceftazidime-avibactam as a first line treatment in ESBL ? Differences between Ceftolozane-tazobactam and Ceftazidime-avibactam should be further developed. A table with spectrum of antimicrobial activity of all the available antibiotics class against ESBL and the different CR enterobacteria is missing. The small paragraph concerning Legionella pneumophila and PJP is not in the scope of this review. 3 paragraph : I would rather see bullet points for each available antibiotics class than this large paragraph A small paragraph concerning the recommended treatment for S. maltophila, Chryseibacterium and E. meningosepticum is missing. The important topics of the optimal treatment duration and whether or not use dual antibiotics should be better developed in the main text rather than in the discussion.
Author Response
Reviewer 2 ----
Although the topic is interesting, it seems to me that the authors should have gone a little bit further into the subject.
Here are a few comments
Title does not reflect the review : I suggest the authors to modify it to Epidemiology, treatment and prevention of nosocomial bacterial pneumonia.
ANS: We thank the reviewer’s suggestion, and revise the title of this revised manuscript accordingly.
There is no mention of the burden of respiratory viruses in nosocomial pneumonia.
ANS: The burden of respiratory viruses in nosocomial pneumonia is very small according to IDSA/ATS 2016 and Europe/Latin America 2017 guidelines. Consequently, we consider that it is not worth mentioning in this manuscript.
“Nosocomial” should be defined.
ANS: We add the definition accordingly.
“MDR P. aeruginosa” should be defined.
ANS: We add this definition accordingly (derived from --- Magiorakos AP et al. Clin Microbiol Infect. 2012 Mar;18(3):268-81. Multidrug-resistant, extensively drug-resistant and pandrug-resistant bacteria: an international expert proposal for interim standard definitions for acquired resistance) in this revised manuscript.
I guess Table 2 reflect probabilistic treatment ? there is no distinction between ESBL at risk patient and CR enterobacteriae. I do not understand what are the different choices of treatment ?
ANS: In Table 2, we have addressed that the choice of antibiotic regimens in the treatment of HAP/VAP is mainly dependent upon the haemodynamic condition of patients, and risk factors of acquiring MDR-pathogens (especially Gram-negative bacteria). In addition, it is indeed difficult to differentiate the risks of acquisition of infections caused between ESBL-producing and carbapenem-resistant Enterobacteriaceae isolates. We add a brief statement in the revised manuscript.
Do the authors propose Ceftolozane-tazobactam and Ceftazidime-avibactam as a first line treatment in ESBL ? Differences between Ceftolozane-tazobactam and Ceftazidime-avibactam should be further developed. A table with spectrum of antimicrobial activity of all the available antibiotics class against ESBL and the different CR enterobacteria is missing.
ANS: Ceftolozane-tazobactam was shown to have in vitro poor activity against SHV-type ESBL-producing K. pneumoniae isolates, hence it will not be the first-line treatment against ESBL producers clinically. By contrast, ceftazidime-avibactam might be an ideal option in the treatment of carbapenemase (KPC, OXA-48-like) and ESBL producers. We do this statement in the revised manuscript.
Nevertheless, after repeat considerations, we would like to apologise that we are not able to create one table recording of all available antibiotic class against the ESBL and different CRE because it is too complicated and difficult to make.
The small paragraph concerning Legionella pneumophila and PJP is not in the scope of this review.
ANS: We delete them accordingly.
3 paragraph : I would rather see bullet points for each available antibiotics class than this large paragraph
ANS: We adjust the text position but preserve most of the original content accordingly.
A small paragraph concerning the recommended treatment for maltophila, Chryseibacterium and E. meningosepticum is missing.
ANS: We revise them accordingly in the revised manuscript.
The important topics of the optimal treatment duration and whether or not use dual antibiotics should be better developed in the main text rather than in the discussion.
ANS: We adjust the context accordingly in the revised manuscript.

Round 2
Reviewer 1 Report
Table 2 includes again Fosfomycin as monotherapy alternative. It must be removed.
Author Response
Table 2 includes again Fosfomycin as monotherapy alternative. It must be removed.
ANS: We deleted it accordingly.
Reviewer 2 Report
No more comment
Author Response
No more comments.
ANS: We thank the reviewer's expertise to make this comment.